# A Collaborative Emergency Drill System for Urban Tunnels Using BIM and an Agent-Based Model

Gang Yu [1,2], Lihua Shi [1,2], Yan Wang [3,*], Jing Xiong [4,5,*] and Yucong Jin [1,2]

1 SILC Business School, Shanghai University, Shanghai 201800, China; jyc_kx@shu.edu.cn (Y.J.)
2 Shanghai University and Shanghai Urban Construction (Group) Corporation Research Center for Building Industrialization, Shanghai University, Shanghai 200072, China
3 Department of Engineering Management, Sichuan College of Architectural Technology, Deyang 610399, China
4 College of Civil Aviation, Nanjing University of Aeronautics and Astronautics, Nanjing 210016, China
5 College of Air Transportation, Shanghai University of Engineering Science, Shanghai 201620, China
* Correspondence: wangyan@scac.edu.cn (Y.W.); crystal_420@126.com (J.X.)

**Abstract:** With the rapid development of smart cities, the refined management of urban highway tunnels has put forward higher requirements for the emergency disposal ability of operation and maintenance personnel. This paper proposed a collaborative emergency drill system for urban tunnels using building information modeling (BIM) and an agent-based model. The objectives of this paper are as follows: (1) To help address the challenge of multi-person collaborative intelligent drills in complex emergency scenarios, this system constructed an emergency collaborative drill model and a virtual emergency scenario description method based on trait-based objects (TBOs). (2) To help address the challenge of the organization and integration of multi-source heterogeneous data in complex emergency scenarios, the system established an emergency scenario generation method through lightweight BIM data, standard emergency plan documents, and virtual emergency scenario data. The system was successfully applied to the Hongmei South Road Tunnel in Shanghai, China. The feasibility of the proposed system provided practical help for tunnel emergency management and was extended to other urban tunnels in Shanghai.

**Keywords:** emergency management of underground space; building information modeling; tunnel emergency drill; agent-based model

## 1. Introduction

The highway tunnel is essential to the public transportation system, and unexpected disasters may occur during operation. To reduce the casualties and property damage caused by disasters, the operation and maintenance personnel will conduct regular drills according to the emergency plan. When a disaster occurs, these drills can help operations staff comply with the standard operating process (SOP) to address emergency accidents, thereby achieving lower losses. Therefore, efficient emergency drills are crucial for improving the operational performance of tunnels. However, due to urban tunnels' complex environment and structures, the disasters and the corresponding treatment procedures are more diverse than in traditional buildings. Consequently, attaining an effective emergency drill for urban tunnels is challenging and meaningful. Many scholars have researched emergency drills, mainly focusing on three parts: emergency management, emergency drill systems based on the BIM (building information model) and VR (virtual reality), and intelligent emergency management.

### 1.1. The Overview of Emergency Management of Underground Space

Regarding emergency management, scholars have mainly focused on formulating an emergency plan and developing an emergency simulation system. In terms of the emergency plan, Merz et al. [1] summarized the natural factors inducing disasters to establish

warning systems. Chen and Li [2] proposed a network-based evaluation model to handle a coordinated control strategy under uncertain events, providing insights for developing a comprehensive emergency management plan. Zhang et al. [3] proposed different safety emergency evacuation requirements for personnel involved in ultra-long tunnels based on the types and severity of risks associated with tunnel operation. Jiang et al. [4] developed a globally optimal evacuation plan based on a coordinated competition mechanism. Wu et al. [5] proposed a risk assessment model for urban public tunnels, which can evaluate the utility of proposed emergency plans.

To overcome the limitations of time and space with field drills, many computer-based emergency drill systems have been developed [6,7]. Meanwhile, many scholars have researched simulation systems to enhance virtual scenarios' realism. Zheng et al. [8] proposed a floor field model to study the dynamic mechanism of fire and smoke spreading affecting pedestrian evacuation. Seike et al. [9] conducted experimental research on tunnel personnel's evacuation speed under different smoke diffusion conditions. Wang et al. [10] developed a PyroSim-based fire simulation model for an underground shopping mall, showing the evacuation distribution analysis of each area.

### 1.2. Emergency Management System Based on BIM and VR

In the AEC (architecture, engineering, and construction) field, BIM and VR have been integrated into the emergency management system to meet the higher requirement of emergency drills in complex architectural structures. Some scholars have used BIM to visualize the details of buildings in virtual fire scenarios, such as escape routes and fire protection facilities [11,12]. They have also utilized VR technology to train individuals in evacuation skills and emergency response capabilities in fire incidents [13,14]. By utilizing these two technologies, emergency drills become more convenient, and scenarios become more diverse, significantly improving the efficiency of fire emergency training. Meanwhile, rational use of BIM information can assist researchers in conducting more comprehensive evaluations of building performance in different scenarios and identifying risk factors. Alizadehsalehi et al. [15] constructed an emergency management system with BIM and unmanned aerial vehicle (UAV) technology to help safety experts identify hazards and formulate appropriate evacuation strategies for safety drills. Li et al. [16] proposed an automatic safety risk identification mechanism based on BIM. Khan et al. [17] developed a soil excavation safety system based on BIM and visual programming language (VPL), which realized the visualization of potential risks. Tang et al. [18] developed an intelligent safety design tool based on depth-first Search (DFS) and BIM technologies, considering various environmental dangers during emergency evacuation. Feng et al. [19] designed a BIM-based indoor positioning framework to support decision-making activities and management tasks at emergency disaster sites.

Moreover, by simulating natural complex scenarios, VR immerses participants in the virtual environment and interacts with elements. The virtual drill scenario combining VR and BIM can distinctly show the interaction between personnel and infrastructure [20], positively increasing realistic training and in turn strengthening the drillers' understanding of the emergency response process under different conditions. Eiris et al. [21] used enhanced 360-degree realistic panorama (PARS) platforms to improve trainees' hazard identification skills. Bin [22] designed a building safety training system, which allows people to learn in the virtual scenario of safety accidents. Ma and Wu [23] constructed a fire emergency management system considering building user behavior decisions, like escape, fire extinguishing, and so on.

### 1.3. Intelligent Emergency Management

Artificial intelligence (AI) technology is being increasingly introduced into the emergency drill system to enhance the efficiency of intelligent emergency management [24]. Guo [25] proposed a standard framework that combines tunnel fire knowledge with machine learning (ML) to provide scientific decision support for intelligent fire protection.

Sharma et al. [26] proposed a hybrid algorithm based on Q-learning and deep Q-network (DQN) to address the fire route evacuation problem. Cheng et al. [27] developed an emergency escape simulation model based on BIM and agent technology, which can accurately simulate personnel evacuation on offshore oil and gas mining platforms. The research guided selecting and optimizing personnel evacuation schemes. Fang et al. [28] proposed a method based on GMM-HMM modeling, which can automatically identify the fire development stage in the residential room and assist drill personnel in making rescue decisions. Ye et al. [29] developed an ML-based model to forecast the outcome of emergency disposal in fire incidents.

In summary, the research in these three aspects has laid a solid foundation for emergency drills in underground tunnels, but some limitations still need to be addressed. Firstly, existing training systems cannot effectively organize and integrate multi-source heterogeneous data in complex emergency scenarios. Although existing methods based on single BIM data can visualize the architectural spatial structure, they cannot interact with heterogeneous data such as emergency plans and emergency behaviors. Research shows that the interaction between BIM and other data in existing virtual scenarios is time consuming and needs standardized processes [30,31]. However, complex emergency drills in underground spaces require combining virtual and natural elements to help drill personnel comprehensively understand the disposal process. Lastly, although some research has used agent technology to simulate actual personnel to address the lack of drilling personnel, they overlook the collaborative cooperation among personnel and only focus on the interaction between personnel and static objects such as buildings. Therefore, the agents simulated by this method cannot interact with actual personnel and cannot achieve whole-process collaborative drills.

To solve these above limitations, this paper proposes a tunnel emergency drill system. The objectives of this paper are as follows: Firstly, to help address the challenge of intelligent drills for multi-person collaboration in complex emergency scenarios, a multi-agent-based model was constructed, which is mainly composed of virtual facility, virtual environment, and drill agent. Next, a virtual emergency scenario description method based on trait-based objects (TBOs) was proposed. To help address the challenge of the organization and integration of multi-source heterogeneous data in complex emergency scenarios, the paper established an emergency scenario generation method based on multi-source heterogeneous emergency data integration through lightweight BIM data, standard emergency plan documents, and virtual emergency scenario data.

## 2. Methodology

The emergency drillers mainly consist of two types: (1) managers primarily located in the control room and providing timely instructions based on the monitored tunnel conditions and (2) field workers primarily responsible for on-site emergency response. The scope of tunnel emergency drills is supposed to cover the following activities: (1) Field workers deploy buffer zones, work zones, downstream transition zones, and termination zones based on the accident situation and place construction signs, like speed limit signs, to alert passing vehicles. They then perform emergency response actions in the accident area, such as vehicle towing and facility restoration. (2) The managers promptly adjust signal lights and broadcast instructions to passing vehicles based on the construction situation. Meanwhile, the managers should provide appropriate supervision and guidance to the field workers' rescue actions based on the emergency response standards. (3) Based on the emergency response standard, the drillers' practical operation ability and theoretical knowledge are evaluated, which helps analyze the drill results comprehensively.

To meet the requirements, a system proposed in this paper was developed using the methodologies described in Sections 2.1–2.4 to construct a collaborative emergency drill system. The system aims to enable emergency drillers to make faster and more accurate decisions in response to unexpected accidents based on predetermined emergency response standards. The system mainly provides the following functions: (1) The system integrates

tunnel structure data into the BIM platform to achieve simulation virtual environments. (2) Considering that field workers have more direct contact with buildings and various accident scenarios, the VR-based module allows field workers to immerse themselves in a virtual environment and interact in real-time with BIM facilities and other emergency drillers to complete emergency response tasks. (3) Because managers need to inspect the actual equipment in the control room, the managers utilize the mixed reality (MR)-based module to achieve real-time third-person perspective monitoring of the emergency drilling process and fulfill specific command tasks based on actual situations. (4) The web-based module enables the visualization of BIM data on the website and the movement path of virtual emergency drillers in virtual scenarios, which provides more diverse data observation perspectives for senior managers who have not participated in the drill. (5) The intelligent multi-user services enhance emergency drilling roles, enabling autonomous completion of emergency training tasks and interaction with emergency drillers. Finally, the emergency drill's efficiency is improved. (6) The data analysis service comprehensively evaluates the drill based on the behavior and records answer data during the drill process.

## 2.1. System Architecture and Workflow

The tunnel emergency drill system designed in this paper adopts the system architecture based on microservice composed of some basic engines, a data layer, a service layer, an access layer, and an application layer from bottom to top, as shown in Figure 1. These basic engines provide essential services, including the server engine, the web application framework, the data engine, the container engine, the monitoring engine, and the security engine. The system implements the deployment of the data layer, the service layer, and the access layer on the basic engine.

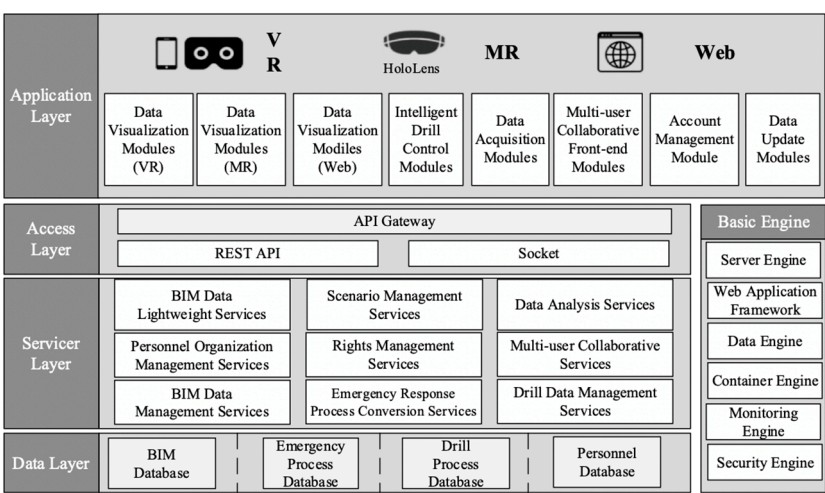

**Figure 1.** System architecture.

The data layer is the cornerstone of the whole system, mainly responsible for the organization and storage of BIM data, emergency process data, deduction data, and personnel data. The service layer provides back-end micro-services, including BIM data lightweight services, scenario management services, data analysis services, personnel organization management services, rights management services, multi-user collaborative services, BIM data management services, emergency disposal process conversion services, and drill data management services. The access layer provides an application programming interface (API) gateway to integrate the interactive interface of microservices. It communicates data with front-end functional modules based on the REST protocol and socket interface.

Among them, the front end and the back end of the multi-user collaborative service needs to share real-time emergency scenario data, so the socket interface is used for data synchronization. Other microservices use the REST protocol to communicate to reduce

system resource consumption and send requests only when data is needed. The application layer is the human-computer interaction channel between operation and maintenance personnel and the system. It is composed of VR, MR, and web clients, including data visualization modules (VR/MR/WEB), intelligent drill control modules, multi-user collaborative front-end modules, data acquisition modules, data update modules, account management modules, etc. The system supports multiple interaction methods such as MR, VR, mobile phones, and tablets to ensure that users under different hardware conditions can have the same drill experience.

The system workflow can be decomposed into three key steps: virtual scenario generation, intelligent drill control, and multi-user collaborative drill, as shown in Figure 2. Firstly, to generate the information required by the virtual scenario, the BIM data is lightweight to extract the geometric and semantic information needed for the virtual scenario. Meanwhile, the unstructured emergency plan document is converted into formatted emergency process data and stored in the database. Then according to the emergency scenario data generation method proposed above, the required information is automatically extracted from the database according to the drill requirements. Thus, scenario data can be dynamically generated. Next, the scenario data visualization service renders the 3D virtual scenario in the clients of different platforms based on the obtained emergency scenario data. Different roles of drillers utilize different clients to control virtual avatar agents. Meanwhile, simulation agents controlled by an intelligent drill engine carry out emergency responses based on predetermined rules. Ultimately, multiple agents work collaboratively to accomplish the emergency drills.

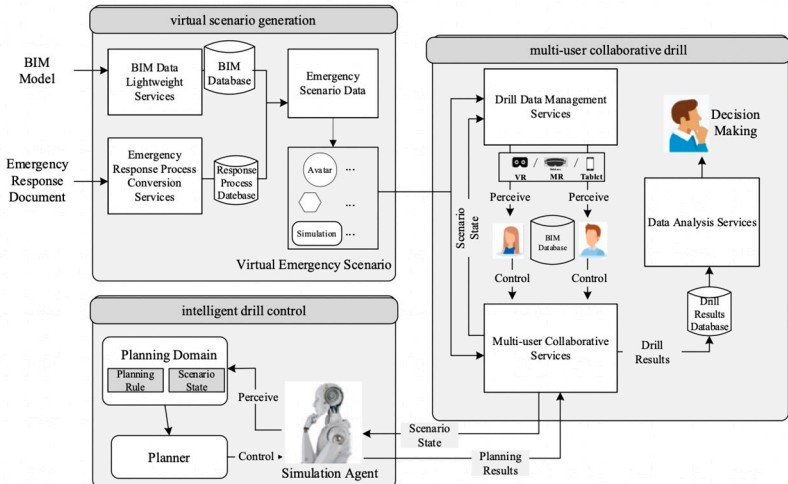

**Figure 2.** System workflow.

### 2.2. Emergency Collaborative Drill Model Based on Multi-Agent

The tunnel emergency collaborative drill needs to be carried out by the operations and maintenance personnel of different roles according to the emergency plan process. However, when drillers are insufficient, the missing roles in the collaborative disposal process cannot be simulated, resulting in the drill not being able to be promoted. To solve this problem, this paper proposed an agent-based emergency collaborative drill model, mainly composed of virtual facilities, a virtual environment, and a drill agent, as shown in Figure 3. Virtual facilities are all kinds of facilities and equipment involved in the emergency plan, which can be operated and controlled by the drilling personnel according to the requirements. The virtual environment simulates various situational objects and elements of the surrounding environment in the actual disaster site. The drill agent is divided into real personnel and virtual personnel. The real driller is mapped to an avatar agent in the model, which is used to obtain the behavior interaction and information perception between the real drillers and the system. The virtual driller is mapped as a simulation agent in the model, which is controlled by the intelligent drill engine designed in Section 2.4. It can constantly perceive

the behavior of real drillers and scenario change information, make decisions according to the existing knowledge or the knowledge obtained by self-learning, operate the virtual facility independently, and cooperate with the real driller to achieve the requirement of an emergency drill. In the case of insufficient drillers for collaborative drilling, the system can automatically generate a simulation agent to simulate all other relevant roles according to the type of drillers required for collaborative drilling.

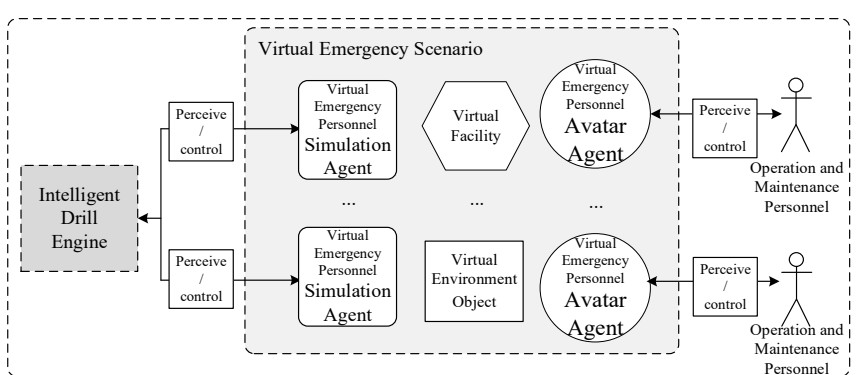

**Figure 3.** Agent-based emergency collaborative drill model.

In order to ensure that the agent perceives the scenario comprehensively, it is essential to standardize the whole process, all personnel and all elements of the real emergency scenario, and establish a virtual scenario suitable for the agent's perception and operation. Therefore, this paper proposed a general description method of the virtual emergency scenario based on trait-based objects (TBOs), as shown in Figure 4.

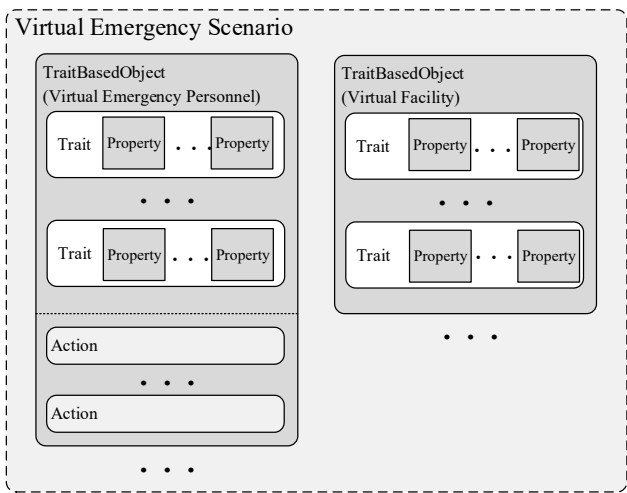

**Figure 4.** The description method of virtual emergency scenario.

Firstly, a virtual emergency scenario (VES) is a limited set composed of TBOs and can be denoted by $VES = \{TBO_1, TBO_2, TBO_3, \ldots, TBO_M\}$, where $M$ is the number of total TBOs in the VES. Among them, $TBO = \langle Traits, Abilities \rangle$. TBO is an independent abstract entity in the virtual scenario, with its own traits and abilities set. Let *Traits* be the set of all traits that a TBO has, and given a trait $i$, let $Trait_i$ represent this trait. A TBO has one or more different traits. These traits are abstracted from the outstanding traits of real objects in emergency plans. The abstraction principle of traits is to merge as many common traits between objects as possible, and there is no dependency between traits. Each trait contains one or more properties. Let $j$ denote the number of properties that a trait has, so a trait is a set of all properties can be represented by $Trait_i = \{property_1, property_2, property_3, \ldots, property_j\}$. The abstract principle of properties is to merge as many common properties between traits as possible and to describe each trait completely with as few properties as possible.

Meanwhile, a TBO may have different actions. Let *Abilities* represent all actions that a TBO has, and given action *n*, let $Action_n$ be this behavior. These actions have different execution trigger conditions, which are called preconditions. When the conditions are satisfied, they can be triggered to execute and produce different effects. Therefore, let $Action_n = \langle Preconditions, Effects \rangle$, where *Preconditions* is the set of all prerequisites for $Action_n$ to execute, and *Effects* is the set of all effects for $Action_n$. The abstract principle of action is to describe preconditions by the logical operation (equal, greater than, less than) of the attribute value of the TBO's traits and to describe the influence by changing, adding, or deleting the TBO's traits. The agents in the emergency collaborative drill model consist of a geometric model and a TBO, as shown in Figure 5. In the virtual emergency scenario, the agent is represented as a 3D geometric model that can be moved, rotated, scaled, and animated.

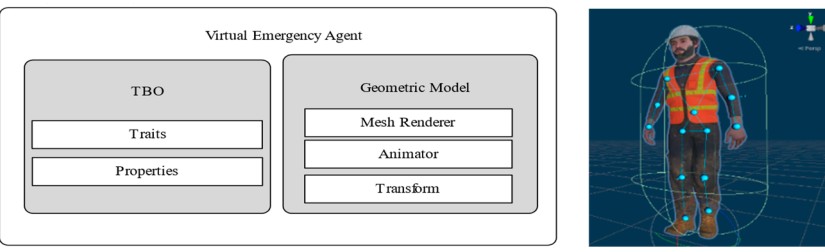

**Figure 5.** The structure and visualization of a virtual emergency agent.

### 2.3. Emergency Scenario Generation Based on Multi-Source Heterogeneous Data

### 2.3.1. Characteristic Analysis and Preprocessing of Emergency Scenario Data

BIM data mainly involves geometric information and logical information of various components. The geometric information is used to describe the 3D coordinates, geometric shape, color, mapping, and other information of the element. Logical information refers to attribute information and associated information of features in BIM data, such as functions, materials, and affiliated relationships of components. Figure 6 shows the geometric and logical information of tunnel BIM data integrated into the emergency scenario. The coding of tunnel facilities and equipment components is a one-to-one correspondence with attribute and correlation information and stored in the database based on MongoDB with a key-value pair structure.

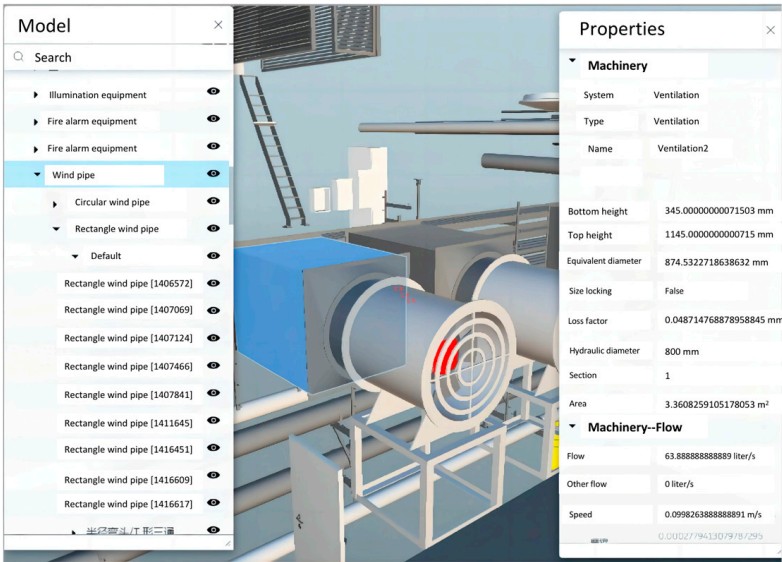

**Figure 6.** The geometric and logical information of BIM.

However, high-precision BIM data will increase servers' and clients' computing and storage pressure, reduce data transmission efficiency, and lead to poor user experiences. Therefore, it is important to lightweight the BIM model's geometric and logical information. The lightweight process is shown in Figure 7. The lightweighting of geometric information focuses on removing vertices and surfaces and reduces the triangular facets without affecting visual perception. In this paper, the edge folding algorithm is used to lightweight geometric information. By calculating the cost required for each edge to be folded and sorting it, the edges are folded from the edge with the least cost until all edges cannot be folded. Then the geometric information is instantiated. Only one of piece the geometric information is retained for the same model construction, and the others are mirrored. For the attribute data corresponding to the BIM model component, the relevant attribute data access is screened according to the needs of emergency drills; irrelevant, redundant data is discarded; and the data is lossless compressed on this basis so as to achieve the purpose of data reduction.

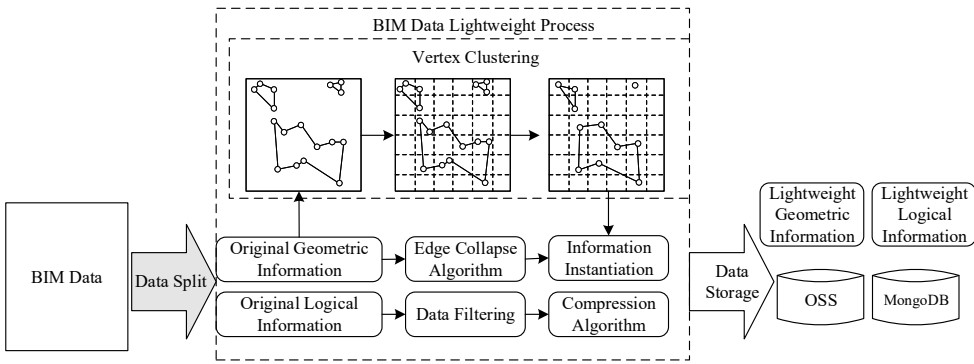

**Figure 7.** The lightweight processing of BIM data.

This paper proposed a standardized conversion process based on the semantic analysis shown in Figure 8 to convert the unstructured traditional emergency disposal process document into structured emergency process data. There are two main steps: (1) The emergency process table is derived from an unstructured emergency plan, which mainly involves five key elements of information, including the following: Emergency State (*STA*), Person in Charge (*PIC*), Emergency Resource (*RES*), Emergency Measures (*MEA*), and Execution Zone (*ZON*). The *STA* refers to the name of the current stage; the *PIC* refers to refers to the role name of the responsible person (such as traffic monitor); the *RES* includes the name (such as speed limit sign), number, and storage location of several types of emergency facilities; the *MEA* includes a detailed description of the specification requirements for the current emergency action; and the *ZON* includes an absolute or relative position of executing the current standard disposal behavior. Next, according to the parsing process of the emergency process table shown in Figure 9, the table is stored in the database. Firstly, based on the hierarchical structure of the emergency process table, the dataset is converted into a tree structure. To begin with, each row of data in the table is traversed, a node is created for each data point, and the parent-child relationship between nodes is determined based on the correspondence between fields. Then all tree nodes are organized according to their parent-child relationship to form a tree structure. Secondly, the tree is traversed using the preorder traversal algorithm, and each sequence in the traversal result is matched with the database field according to the tree level. Finally, complete emergency disposal process data is imported into the database. (2) According to the user's drill scenario requirements, the standard emergency response actions of different personnel in various states are dynamically extracted. This process generates the desired emergency response process data for subsequent evaluation purposes.

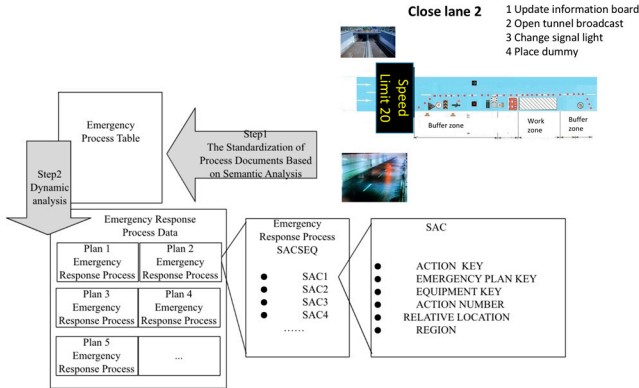

**Figure 8.** The standardized conversion process of emergency disposal process data.

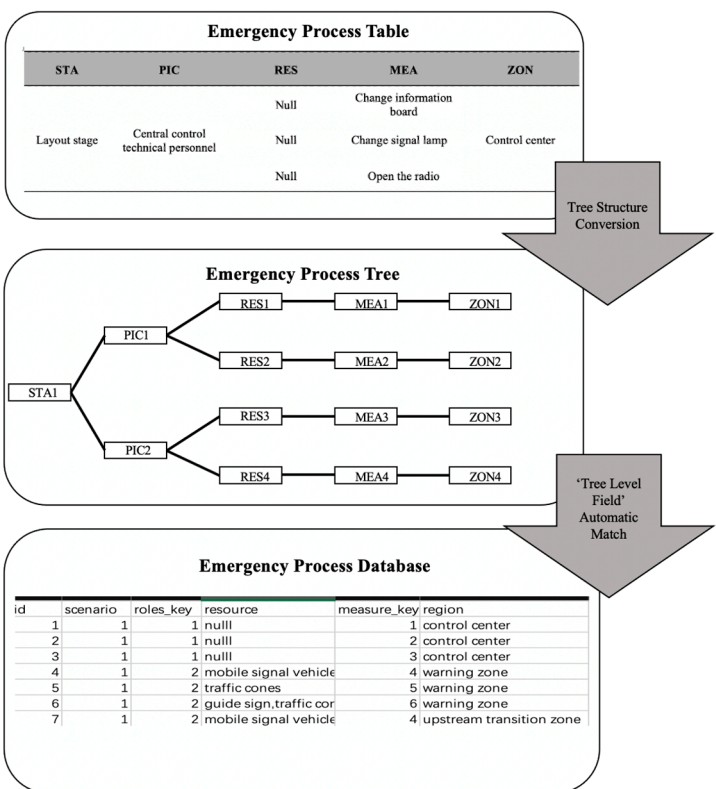

**Figure 9.** The parsing process of the emergency process table.

### 2.3.2. Dynamic Generation and Visualization of Emergency Scenario Data

Based on the above data, the generation of emergency scenario data is firstly based on BIM data, and the basic emergency scenario information is constructed according to the requirements of user-defined drill scenarios. Secondly, the emergency disposal process is automatically and dynamically extracted from the emergency process data to generate emergency drill evaluation information, which mainly includes standard disposal specifications, evaluation indicators, evaluation principles, and so on. It provides the basis for evaluating the effect of this drill. Then a scenario is generated related to the TBO set for recording all the changes in the TBO's behaviors and traits in the drilling process to form scenario drill data. The generation process of emergency scenario data is shown in Figure 10. When the emergency scenario data are generated, the prefab technology of Unity is used to traverse the emergency drill evaluation information and create an assessment prefab, which is instantiated in the virtual scenario. Next, the TBO set is analyzed according to the virtual emergency scenario description method introduced in Section 2.2 to generate

a TBO prefab. Meanwhile, the position, angle, and traits of this prefab are set, and then the prefab is instantiated in the scenario. At the same time, according to the basic information of the scenario, the corresponding BIM data is loaded into the engine to render and visualize the emergency scenario. The above visualization process of emergency scenario data is shown in Figure 11.

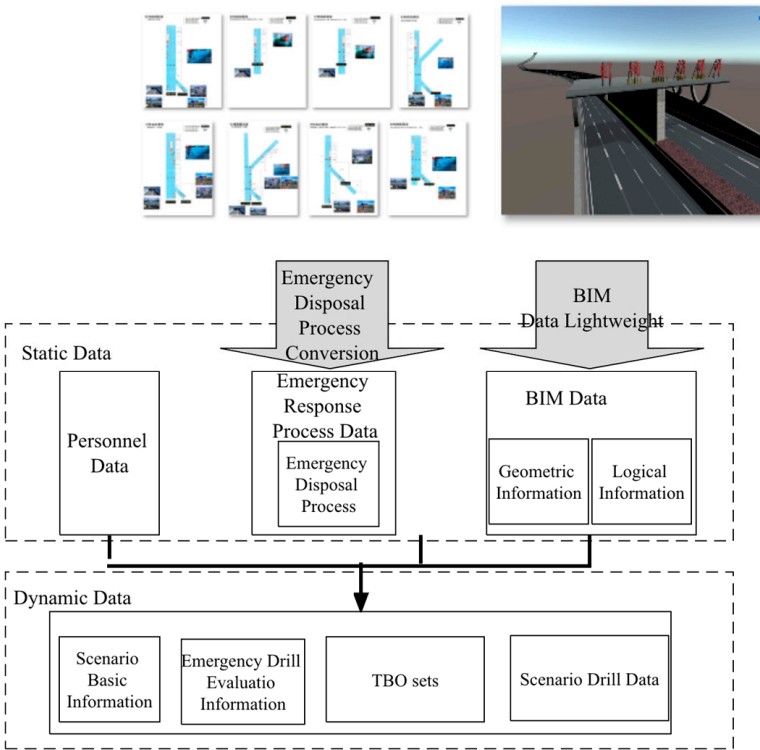

**Figure 10.** The generation process of emergency scenario data.

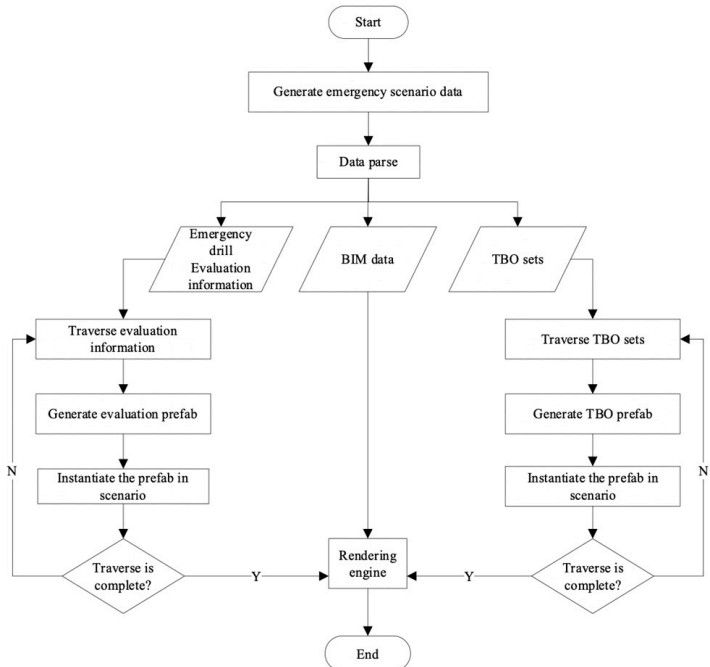

**Figure 11.** The visualization process of emergency scenario data.

### 2.4. Intelligent Drill Engine Based on Multi-Agent

The control of the agent in the virtual scenario is realized by the intelligent drill engine, which mainly includes a planning domain and a planner. The planning domain (PD), $PD = \langle PlanningRule, VES_t \rangle$, is used to store the planning rules and the scenario state at the current time $t$. In addition, the *PlanningRule* consists of *TraitCollection*, *PropertyCollection*, *ActionCollection*, and *TerminationCollection*, which refers to the rules for the simulation agent running in the scenario. In particular, there can be more than one termination condition. When the scenario state $VES_t$ satisfies the condition, the planning stops. The PD generates the corresponding behavior tree according to the planning rules and functions of different agents. For example, the main actions of the traffic monitor include confirming whether a fire occurs, changing the signal lights in the facility, updating the information board message, and opening the tunnel evacuation broadcast. The corresponding simulation agent behavior tree structure is shown in Figure 12.

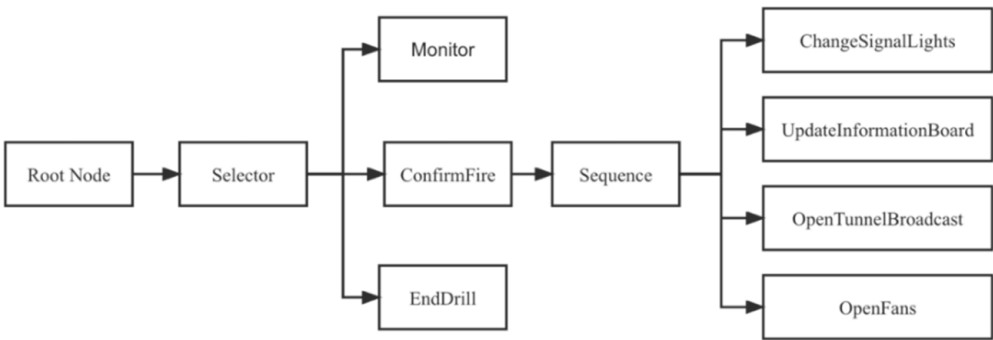

**Figure 12.** Behavior tree structure of the simulation agent.

Planner. In the emergency collaborative drill process, the planner obtains the planning rule and status the $VES_t$ from the PD. Then the output planning result is returned to the simulation agent in the virtual emergency scenario. In this paper, the behavior tree and the A-STAR algorithm are used to guide the action of the simulation agent to ensure that the planning result output by the planner can drive the agent to complete the drill. In the planner, the behavior tree nodes are traversed by depth-first execution to realize the behavior's simulation planning. As a result, the simulation agent can select the appropriate behavior by combining different scenarios in the plan with the current state. At the same time, if the position change of the simulation agent occurs in the planning, the A-STAR algorithm is used for path planning to generate new position information [32]. The A-STAR algorithm can automatically schedule the optimal action path from the current state to the target position for the agent, which can efficiently and accurately complete the emergency collaborative drill. The process of the emergency drill path planning based on the A-STAR algorithm is shown in Figure 13.

In the emergency collaborative drill, the logic of the engine controlling the simulation agent is as follows: Let the state of the virtual emergency scenario at time $t$ be $VES_t = \{TBO_j | j = 1, 2, 3, \ldots, k\} |$. After sensing $VES_t$, the intelligent drill engine performs action planning to obtain a planning result $R = \{Action | l = 1, 2, 3, \ldots, m\}$. $R$ is the set of next actions of multiple simulation agents. After obtaining the planning result, the agents control the geometric model and the TBO for animation playback and action execution, respectively. When the planning results of all agents are completed, the virtual emergency scenario state changes from $VES_t$ to $VES_{t+1}$. Repeat the above steps until the drill ends. This control logic is shown in Figure 14.

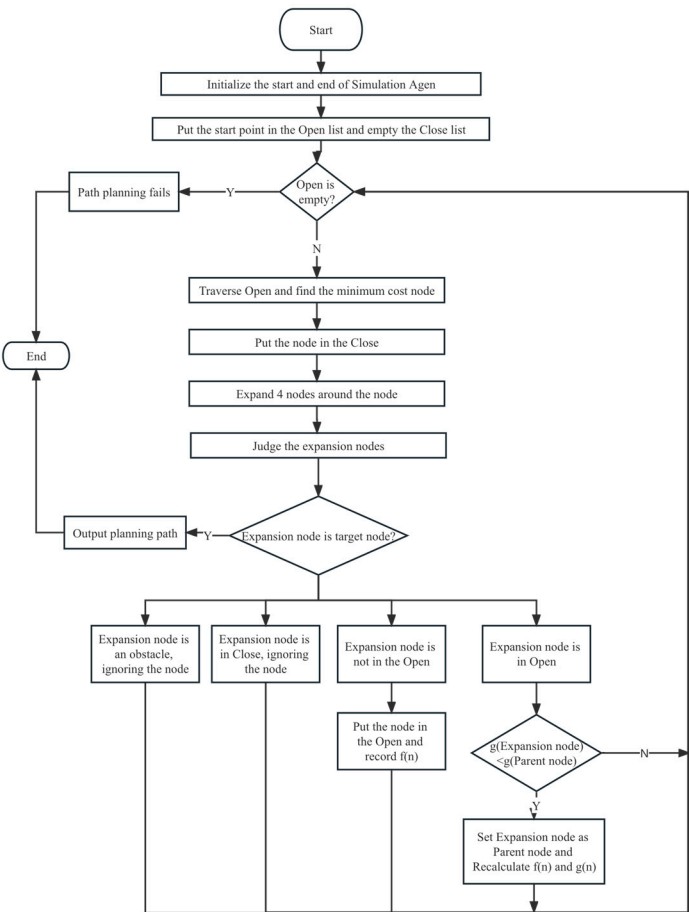

**Figure 13.** The process of emergency drill path planning based on A-STAR algorithm.

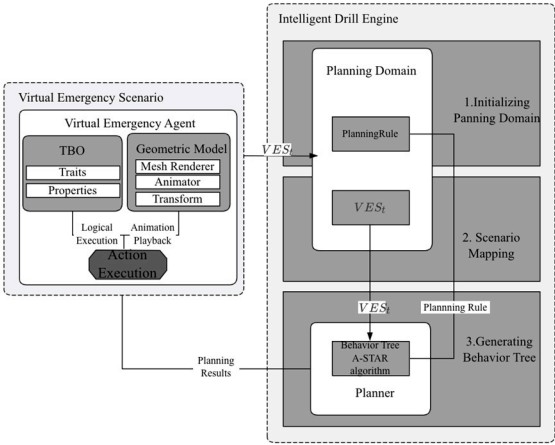

**Figure 14.** The control logic of the simulation agent.

## 3. Case Study

Hongmei South Road Tunnel is located in Shanghai, China, with a total length of 5.26 km. In addition to the traffic layer, the tunnel has another layer on top and bottom. The upper layer is mainly used for exhaust air, and the lower layer has ample space for emergency use. The emergency corridor is divided into three areas. The middle area is dedicated to fire trucks, the two sides are used for pipeline maintenance and evacuation, and there is a location mark every 200 m. To improve the efficiency of operation and maintenance and reduce the risk of accidents, tunnel operation and maintenance enterprises introduced the system proposed in this paper for emergency drills.

### 3.1. Implementation of System

According to the system framework described above, the system chooses a cloud server as the platform carrying the service layer and data layer, Windows Server 2012 as the server operating system, Docker as the microservice container engine, and Node.js as the backend microservice development framework. The deployed databases include SQL Server-based relational databases, MongoDB-based key-value databases, and OSS (Object Storage System)-based binary file data storage systems. The microservice implements data communication with the database based on the port provided by the database engine. The MR client runs on Microsoft HoloLens, the mobile VR client is carried on the Android tablet, and the client development tools use Unity engine, Visual Studio, and Mixed Reality Toolkit. Figure 15 shows the HoloLens and Android tablets equipped with client software. The web client is developed based on the React16.8.6 front-end framework, uses AntDesign3.20.0 components to build the user interface, and uses Axios0.19.0 to achieve front-end and back-end communication based on the REST protocol.

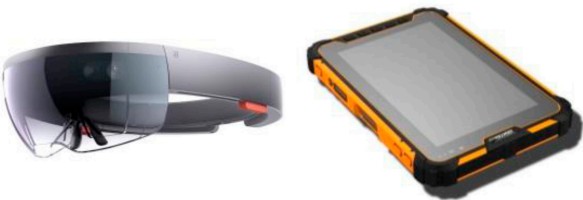

**Figure 15.** HoloLens and Android tablet.

### 3.2. Intelligent Drill Control

In the drilling process, the simulation agent is driven by the engine, which can make decisions and execute actions autonomously in various virtual emergency scenarios and complete the drill in collaboration with the avatar agent controlled by the actual drill personnel. Figure 16 demonstrates the simulation agent driven by the engine in the central control room scenario, the avatar agent controlled by the real driller in the tunnel scenario, and the collaborative drill between the simulation agent and the avatar agent. As shown, when there are few drill personnel, the simulation agent can independently carry out emergency responses according to the predetermined process and interact with the avatar agent to complete drill tasks together.

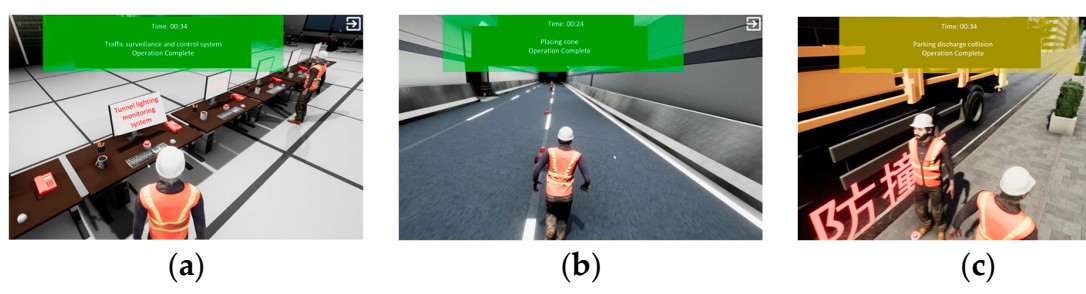

| (**a**) | (**b**) | (**c**) |

**Figure 16.** The realization of intelligent drill engine (**a**) simulation agent; (**b**) avatar agent; and (**c**) collaborative drill.

### 3.3. Multi-User Collaborative Drill

Before the multi-user collaborative drill starts, users need to create a host for the collaborative drill. Other users can join the host according to the host's IP address to complete the drill together. When the number of users does not meet the number required for the drill, the user can also start a collaborative drill. In this case, the system will automatically generate a simulation agent to complete the drill with the user. In addition, the system will automatically demonstrate the plan knowledge, such as the plan processing flowchart and plan core processing video. Operation and maintenance personnel can use this to understand the flow of the entire plan and focus on learning core disposal tasks. The

system not only examines the practical operation skills but also examines the understanding of theoretical knowledge. Figure 17 shows the initial system configuration interface, the screen shots of the disposal process, and a video demonstration.

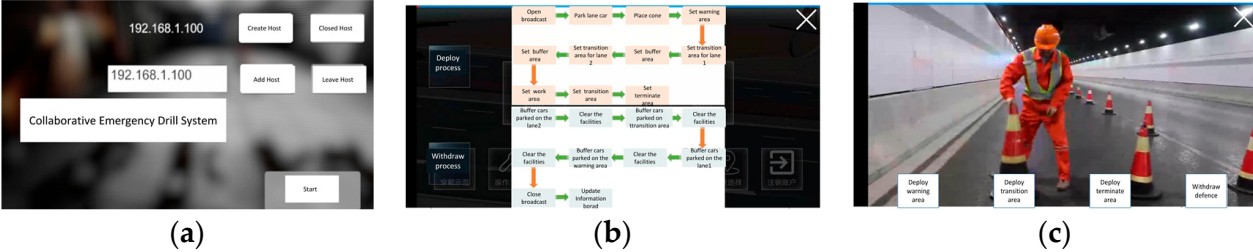

**Figure 17.** System operation interface (**a**) initial system interface; (**b**) disposal process; and (**c**) video demonstration.

Operation and maintenance personnel operate the avatar agent and complete emergency disposal tasks per the plan's process. According to the fire emergency plan, it is necessary to confirm the fire in the tunnel control room; inform the fire department, traffic police, and health care departments; open the tunnel broadcast; and open the fan. In addition, it is supposed to park anti-collision vehicles and place road cones, warning signs, and simulators at the tunnel site. The drillers need to perform corresponding emergency disposal operations according to different virtual emergency scenarios. At the meantime, during the drill, the system automatically generates interactive questions and answers on crucial disposal tasks according to historical drill records, which are used to evaluate drillers' mastery of emergency disposal standards. Figure 18 exhibits the confirmation operation of the agent in the central control room, the emergency drill process screenshots of MR and other clients, and the screenshots of answering the problem.

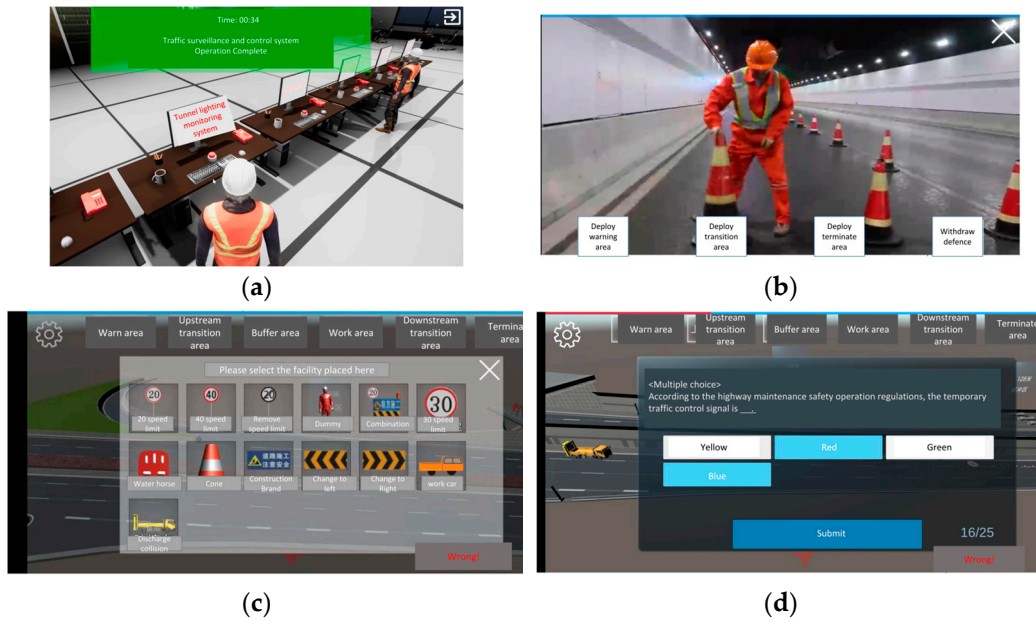

**Figure 18.** The implementation interface of emergency drill (**a**) confirmation operation of central control room; (**b**) drill process of MR clients; (**c**) drill process of other clients; and (**d**) answering the problem of emergency specification.

### 3.4. Evaluation and Discussion

The system has been practiced in Hongmei South Road Tunnel since May 2019. Operation and maintenance personnel can conduct drills anytime and anywhere through devices such as Android tablets and HoloLens and complete the drill plan arranged by the managers.

This study adopts a questionnaire to assess the effectiveness and usability of the designed system. The first three questions are related to the subjects' age, level of management, and familiarity with BIM and VR/MR. After answering these questions, the participants are asked to offer their level of agreement with the statements shown in Table 1 using a seven-point Likert scale. These questions are adapted from the IBM Post-Study System Usability Questionnaire (PSSUQ) [33]. The questionnaire primarily evaluates the system's usability and satisfaction across four sub-scales: system usefulness (Q1–Q6), information quality (Q7–Q12), interface quality (Q13–Q15), and overall (Q1–Q16). The overall score is calculated by averaging the scores from the seven points of the scale. The optimal standard was derived from extensive historical research on computer application systems. Generally, scores below the benchmark indicate higher system performance and good user usability.

**Table 1.** PSSUQ questionnaire.

| Question Number | Question | Answer |
|---|---|---|
| Q1 | Overall, I am satisfied with how easy it is to use this system. | 1 = Strongly Agree to 7 = Strongly Disagree |
| Q2 | It was simple to use this system. | 1 = Strongly Agree to 7 = Strongly Disagree |
| Q3 | I was able to complete the tasks and scenarios quickly using this system. | 1 = Strongly Agree to 7 = Strongly Disagree |
| Q4 | I felt comfortable using this system. | 1 = Strongly Agree to 7 = Strongly Disagree |
| Q5 | It was easy to learn to use this system. | 1 = Strongly Agree to 7 = Strongly Disagree |
| Q6 | I believe I could become productive quickly using this system. | 1 = Strongly Agree to 7 = Strongly Disagree |
| Q7 | The system gave error messages that clearly told me how to fix problems. | 1 = Strongly Agree to 7= Strongly Disagree |
| Q8 | Whenever I made a mistake using the system, I could recover easily and quickly. | 1 = Strongly Agree to 7 = Strongly Disagree |
| Q9 | The information (such as online help, on-screen messages, and other documentation) provided with this system was clear. | 1 = Strongly Agree to 7 = Strongly Disagree |
| Q10 | It was easy to find the information I needed. | 1 = Strongly Agree to 7 = Strongly Disagree |
| Q11 | The information was effective in helping me complete the tasks and scenarios. | 1 = Strongly Agree to 7 = Strongly Disagree |
| Q12 | The organization of information on the system screens was clear. | 1 = Strongly Agree to 7 = Strongly Disagree |
| Q13 | The interface of this system was pleasant. | 1 = Strongly Agree to 7= Strongly Disagree |
| Q14 | I liked using the interface of this system. | 1 = Strongly Agree to 7 = Strongly Disagree |
| Q15 | This system has all the functions and capabilities I expect it to have. | 1 = Strongly Agree to 7 = Strongly Disagree |
| Q16 | Overall, I am satisfied with this system. | 1 = Strongly Agree to 7= Strongly Disagree |

Thirty participants participated in the test. Table 2 summarizes the participants' collected basic information. Most subjects (73.3%) are aged between 25 and 40. The majority of subjects (80%) are workers who need regular security training to ensure that they can take appropriate emergency response actions. Furthermore, 53.3% of the participants used VR, and 60% used BIM in their projects. Tables 3 and 4 present the statistical results. The results indicate that users' ratings for the system are below the benchmark scores, suggesting a high level of user satisfaction with the system. The user's low evaluation of information quality indicates that there is room for improvement in the system's presentation and selective display of multi-source data. On the other hand, the survey respondents expressed high satisfaction with the system's interactive interface, suggesting that the current display pages are effective.

**Table 2.** Basic information of participants.

| Variables | | Frequency | Percentage (%) |
|---|---|---|---|
| | 18–24 | 3 | 10 |
| Age | 25–40 | 22 | 73.3 |
| | 41–55 | 5 | 16.7 |
| Level of management | Worker | 24 | 80 |
| | Manager | 6 | 20 |
| VR/MR Experience | Yes | 16 | 53.3 |
| | | 14 | 46.7 |
| BIM Experience | Yes | 18 | 60 |
| | | 12 | 40 |

**Table 3.** Results of questionnaire.

| | Results | | | Standard | |
|---|---|---|---|---|---|
| Question Number | Mean | SD | 99% Confidence Intervals | Mean | 99% Confidence Intervals |
| Q1 | 1.87 | 0.19 | 1.34–2.39 | 2.85 | 3.09–2.60 |
| Q2 | 1.83 | 0.16 | 1.39–2.27 | 2.69 | 2.93–2.45 |
| Q3 | 1.77 | 0.16 | 1.33–2.2 | 3.16 | 3.45–2.86 |
| Q4 | 1.57 | 0.16 | 1.13–2 | 2.66 | 2.91–2.40 |
| Q5 | 1.63 | 0.16 | 1.19–2.08 | 2.27 | 2.48–2.07 |
| Q6 | 1.67 | 0.18 | 1.17–2.17 | 2.86 | 3.17–2.54 |
| Q7 | 1.97 | 0.20 | 1.41–2.52 | 3.7 | 4.05–3.36 |
| Q8 | 2.07 | 0.15 | 1.65–2.48 | 3.21 | 3.49–2.93 |
| Q9 | 1.80 | 0.17 | 1.33–2.27 | 2.96 | 3.27–2.65 |
| Q10 | 1.63 | 0.12 | 1.3–1.97 | 3.09 | 3.38–2.79 |
| Q11 | 1.77 | 0.19 | 1.24–2.29 | 2.74 | 3.01–2.46 |
| Q12 | 1.63 | 0.16 | 1.21–2.06 | 2.66 | 2.92–2.41 |
| Q13 | 1.70 | 0.15 | 1.28–2.12 | 2.28 | 2.49–2.06 |
| Q14 | 1.47 | 0.11 | 1.15–1.78 | 2.42 | 2.66–2.18 |
| Q15 | 1.47 | 0.11 | 1.15–1.78 | 2.79 | 3.07–2.51 |
| Q16 | 1.33 | 0.09 | 1.09–1.57 | 2.82 | 3.09–2.55 |

**Table 4.** Results of sub-scales.

| | | Results | | | Standard | |
|---|---|---|---|---|---|---|
| Sub-Scales | Included Question | Mean | SD | 99% Confidence Intervals | Mean | 99% Confidence Intervals |
| System Usefulness | Q1–Q6 | 1.72 | 0.10 | 1.46–1.99 | 2.8 | 3.02–2.57 |
| Information Quality | Q7–Q12 | 1.81 | 0.09 | 1.56–2.06 | 3.02 | 3.24–2.79 |
| Interface Quality | Q13–Q15 | 1.54 | 0.08 | 1.32–1.77 | 2.49 | 2.71–2.28 |
| Overall | Q1–Q16 | 1.70 | 0.07 | 1.52–1.88 | 2.82 | 3.02–2.62 |

The results of daily drills will be automatically entered into the database. The administrator can set up a targeted drill plan according to each employee's drill situation to help emergency personnel comprehensively improve their emergency disposal capabilities. The system automatically determines whether the user's action behavior is correct according to the provisions of the emergency plan. The overall drill accuracy is the average of the answer score and the operation score. Meanwhile, the answer score reflects the degree of drillers' theoretical knowledge, which can be defined as $AnswerScore = 100 - 5 \times (n_{TF} + n_{solo} + n_{multi})$, where $n_{TF}$ is the total number of incorrect judgments, $n_{solo}$ is the total number of incorrect responses for multiple-choice questions,

and $n_{multi}$ is the total number of incorrect responses for multiple-response questions. Additionally, $OperationScore = 100 - 10 \times n_{wrong}$ where $n_{wrong}$ is the total number of incorrect emergency response actions, which reflects practical ability.

Starting from May 2019, the Hongmei South Road Tunnel has carried out a monthly simulation drill for 7 consecutive months using two different methods: traditional field drills and the proposed system. The number of participants in each method accounts for more than 90% of the total number of operation and maintenance personnel. Table 5 shows the effect comparison of the drill between the personnel with and without the system. Initially, the results of drills using traditional methods and the proposed system were similar, with low scores. It indicates that both operation and maintenance personnel groups could not handle emergency situations in the tunnel correctly. However, after multiple drills, the drillers using the proposed system achieved higher scores and showed more significant improvement. This validates the effectiveness of the system in enhancing emergency drill performance.

**Table 5.** Comparison results of the drill with and without the proposed system.

| Time | Drill Accuracy | |
| --- | --- | --- |
| | Using the New System | Using Traditional Drill Method |
| May 2019 | 47.5 | 50 |
| June 2019 | 67.5 | 55.5 |
| July 2019 | 67.5 | 60 |
| August 2019 | 80 | 67.5 |
| October 2019 | 77.5 | 70.5 |
| November 2019 | 80 | 75 |

## 4. Conclusions

This paper proposed a tunnel emergency drill system based on microservice, which integrates BIM and an agent-based model. Compared with the traditional drill mode, the system has the following characteristics: (1) Based on BIM, emergency process documents and other multi-data were introduced, and an emergency scenario generation method based on multi-source heterogeneous data integration was proposed. (2) An emergency collaborative drill model based on an intelligent agent was constructed. Co-training was realized by stimulation agents with autonomous decision-making functions so that a multi-person collaborative drill is no longer limited by time and space, drillers, and scenarios. Finally, this paper proved the feasibility and practicability of the system through practical application cases. (3) A system based on microservice was designed, which realizes the rapid transmission of multi-source heterogeneous data and the cross-platform human-computer interaction based on VR/MR.

The application case showed that the system can generate a simulation agent to collaboratively assist with drill tasks based on an intelligent drill engine when personnel are scarce. The system makes the drill more flexible and can be conducted without gathering all department staff. At the same time, the combination of BIM and VR/MR technology allows users to immerse themselves more in virtual scenes and have a more comprehensive understanding of emergency processes. Moreover, a drill system based on multi-source heterogeneous information can better assess personnel's understanding of emergency processes from both practical operation and theoretical knowledge. Finally, the increase in drill scores also proves that the system can improve the efficiency of emergency drills. Consequently, the proposed system is practical and effective.

However, this article still has some limitations. The intelligent drill engine can only simulate exercise behavior based on predetermined scenarios. This means that when unexpected scenarios occur, the simulation results of the engine will be invalid. In the future, reinforcement learning can be considered to enable the intelligent body to handle more intelligent exercises in complex situations. At the same time, the model can be extended to the coverage area of a vast area network using low latency and high

reliability 5G communication technology, achieving remote multi-person collaborative emergency drills.

**Author Contributions:** Conceptualization, G.Y.; methodology, G.Y. and Y.W.; software, L.S. and J.X.; validation, L.S., Y.W. and J.X.; writing—original draft preparation, G.Y. and L.S.; writing—review and editing, J.X. and Y.J.; supervision, G.Y. All authors have read and agreed to the published version of the manuscript.

**Funding:** This research was sponsored by Shanghai Municipal Science and Technology Commission (No. 18DZ1205502. No. 20DZ2251900) and Shanghai Municipal Natural Science Foundation (No. 21ZR1423800).

**Institutional Review Board Statement:** Not applicable.

**Informed Consent Statement:** Not applicable.

**Data Availability Statement:** Not applicable.

**Conflicts of Interest:** The authors declare no conflict of interest.

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
