# Peer review of "A Collaborative Emergency Drill System for Urban Tunnels Using BIM and an Agent-Based Model"

_sustainability, doi:10.3390/su151813533_

Round 1

Reviewer 1 Report

1. First of all, thank you for integrating artificial intelligence into building information modeling and constructing a framework for a multi-person, full-field collaborative intelligent exercise system in tunnel scenarios based on mixed reality technology. The overall structure of the article is complete, the workload is high, and it has achieved good application results, which is worthy of recognition. However, some of the content and diagrams in the article still contain certain errors, and corrections are recommended. 

2. It is recommended that chapters one and two be combined, that the literature review be restructured so that it does not need to be presented in two chapters, and that the content be appropriately trimmed and combined.

3. The title of this article is the framework of the emergency exercise system, which is large in scope, while the methodology in the article focuses mainly on the management and presentation of data and the behavioral planning and control of intelligence, and the framework structure of the system is only briefly described in the case study. These lead to an overly brief introduction of each part, and it is suggested that the selection be suitably narrowed down to focus on the improvement of the analysis methodology, with appropriate detail.

4. The text in most of the images in the article is too small and difficult to read, and a small number of images are very blurred; Figure 11 is particularly odd, with a stitched-together feel, and it is recommended that they be redrawn and inserted. Typographical errors in lines 372-382 of the article.

5. The data normalization conversion process and text description in Figure 6 do not explain how the tables are converted into a tree structure and how the tree levels are defined. It is recommended that an explanation be added accordingly and that each conversion step be made more specific as to how it works.

6. What is the A* algorithm mentioned in section 3.3? Suggest adding the appropriate explanation.

7. Please note that abbreviations in articles should be given in full the first time they are used, e.g. AEC.

1. The whole sentence of the article is smooth, but there are some problems with sentence structure, verb tense, and clause structure.

2. You need to pay particular attention to English grammar, spelling, and sentence structure so that the goals and results of the study are clear to the reader.

Author Response

We greatly appreciate the thorough and thoughtful evaluation of our paper by the anonymous reviewers.  Their insightful comments helped us significantly improve the quality of this paper.

Following your advice, we have undertaken a minor revision of this manuscript. In the files of revision with changes marked, the yellow highlights are changes for responses to the editor and reviewer comments.

Reviewer 2 Report

There are several problems:

* The title promises something which is not to be found in the article
The first part of the title states "Integrating Artificial Intelligence into Building Information Modeling". But there is no proof in the paper that AI is integrated into BIM. It seems that both technologies are used but do not interact. As presented in the paper, BIM is used for the 3D model and this 3D model is used for MR. And AI is used for the emergency management (Section 2.3). But it is never explained how AI interacts and changes the BIM model. In Section 3.2 it is stated that the relevant attribute data is screened according the needs of emergency drill. But it is not clear why AI should be necessary for that screening.
This is much better described in the abstract: "This paper proposed a new framework for an emergency drill system based on Mixed Reality (MR), which integrates Building Information System (BIM) and Artificial Intelligence(AI)." 

* Structure of the article (Are the research design, questions, hypotheses and methods clearly stated?)
The paper describes an interesting development: an tunnel emergency drill system based on mixed reality with all the technology (BIM, AI). All the components are described. But there is no research design described (except using the different technologies), the research question is missing as well as any hypotheses. The methodology section (Section 3) does not describe the research methodology; instead it describes the technologies used.
Then a case study is presented. Most of the article, the authors talk about the use of VR. In the case study, then the use of the Microsoft HoloLens is mentioned. The HoloLens is an Augmented Reality HMD (head mounted display). It is not clear why the HoloLens is used for the MR client, because it overlays the virtual elements in the real environment.

* Intelligent drill engine based on AI unclear
Section 3.3 describes the intelligent drill engine based on AI. After reading this section it is unclear why AI is necessary. AI is best when it can be trained with millions of data sets. It is not clear how this AI is trained. It is not clear if a very good algorithm is used instead of AI.

* Discussion of research (Are the arguments and discussion of findings coherenct, balanced and compelling? For empirical research, are the results clearly presented?)
The is no discussion of the research results. The case study describes the use of all the elements of the developed system. But there is no discussion.

* Conclusions of research (Are the conclusions thoroughly supported by the results in the article or referenced in secondary literature?)
The conclusion section is very short (2 paragraphs, 25 lines). Some conclusions are right: a new model was proposed, constructed, and designed. But this is more a summary.
The conclusion that the paper "proved the feasibility and practicability" is not shown in the paper. 
Another conclusion states the "proposed ... framework achieves a better drill experience and more convenient use, which significantly improves the effect of emergency drills by breaking through the limitation of drillers and space."
All theses conclusions must be shown by means of experiments, but these experiments are missing in the paper.

* Possible unnecessary table and figure
It is not clear why Table 1 is necessary. The information can be written in a sentence.
Figure 18 has no important information.

Extensive editing of English language required?
Several sentences are much too long which also sometimes leads to a wrong grammar or wrong time (past tense instead of present tence and vice versa).

Style:
Many times spaces are missing after a dot.
The lines 260 to 289 are not formatted in justification.
Figure 9 is located between the text and the caption of this figure is not below the figure but many lines above and also located between the text.

Author Response

(The authors gave the same response as above.)

Reviewer 3 Report

The article is devoted to a very important issue - a new framework for an emergency drill system based  on Mixed Reality , which integrates Building Information Modeling and Artificial Intelligence.

At the same time, the proposed solution integrates several Industry 4.0 technologies at once.

In general, the article is well written, the approaches used are described in detail. There are no questions about the content of the article. However, there are several suggestions for improving the article.

Source 20 dates from 1994, given that the article is about the use of modern technology, it is advisable to update the source.

Picture title 9 moved out (line 384).

In addition, it is advisable to present an economic calculation for the implementation of the proposed solution.

The submitted article is well written.

Author Response

(The authors gave the same response as above.)

Round 2

Reviewer 2 Report

General comments

The paper has still errors and shortcomings that should not be in a research paper. So far, it is still more of a report than a research paper.

Authors' response the the reviewer comments

The authors did not address all comments, even omitted some important ones (concerning structure, discussion, conclusion).

Response to prior comments concerning MR:
The authors mention changes in line 245-257 or Figure 7.
Neither the text nor the figure has anything to do with MR.

Response to prior comments abouth AI:
The authors mention that they don't need historical data to train the AI.  That's fine, but then they still need to explain what kind of AI is used, how the AI is used, and how it is an integral part of their research.
A* is not an AI as far as I know, but a pathfinding algorithm (which does not use AI).

Response to prior comments about conclusion:
Lines 412-414: This is okay, but it does not depend on the number of drilling personnel present (it also works if there are a lot of drilling personnel present).
Lines 427-429: This statement doesn't say a lot. Why should the system "examine" the practical operation skills and the theoretical knowledge?
Lines 485-495: This statement is not supported in the text before. 

Abbreviations

When using abbreviations, it is important to use the correct expression the first time it is used and to describe the expression.

Title

The title again uses buzzwords: multi-source heterogenous data and artificial intelligence. But then it would be important to justify the use of these words in the text and describe exactly why these words are used and how they are an important part of the research.
It is completely unclear how "multi-source heterogenous data integration" is actually used. Emergency plans and emergency behavior are a bit little for "heterogenous data". Then, every BIM model would be "heterogenous data".
It is completely unclear where and how AI is used ... and how the used "AI" is self-learning.

Methodology

There is no explanation of the content and scope of a drill.
The requirements for the system are completely missing.
Later in the case study (lines 436-441), some requirements are described, but they need to be described and extended in the methodology. Otherwise it is not clear what the newly developed system is being compared to.
Evaluation criteria are completely missing. 

Use of AI

Still it is not described or reflected in the work how AI was used.
A* is not an AI as far as I know, but a pathfinding algorithm (which does not use AI).
There is no prove presented that there is a self-learning process of "AI" which is important for an AI. 

Objectives of the paper missing

It is important to describe the objectives of the research. Then these objectives can be evaluated.

Results and evaluation

Results and evaluation are missing!

In the very short subsection 3.5. (Application), the authors have include a short paragraph. Figure 19 does not have any surplus value. It is not explained why this picture is important. This is unnecessary.
Figure 20 is just mentioned in one sentence ("Figure 20 shows the line chart of the average score of 7 monthly assessments, including operation score, answer score, and total score."). BUT: these scores are never described. It is unclear exactly what these scores are, how they are measured, what the highest point value of these scores is, and why they are important for the process and the new system.

Also Figure 21 is only mentioned in one senctence.

Since neither the scoring system nor the importance of the scoring system is described the statement in lines 466-468 (improvement of knowledge mastery and operational proficiency) is not clear.

In general, figures have to be described better.

It is logical that personnel will get better over time. But it is not clear whether the personnel get better BECAUSE they used the new system - there is no prove in the paper. 

Discussion of research

There is still no discussion of research results.

Conclusions

The conclusion section is still not supported by the text of the paper.
It has not improved to the too short conclusions of the first version of the paper.

All conclusions must be shown by means of experiments, but these experiments are still missing or not described in the required extent.

The conclusions prominently mention "AI technologies" and "based on artificial intelligence", but these "AI technologies" are not explained in the text. 

References

Interestingly, the authors have now used fewer references than in the first version.

They use terms at the end of the titles in each reference which are unclear (e.g., [J], [C] ... maybe Journal paper and Conference paper ... but this is dispensible). All doi numbers are missing.

In reference [31], the authors are not correctly indicated and, in addition, "ScienceDirect" is listed instead of the journal alone:
ASTM, ADA, AAB, AMG, BRM. Building information modeling for facilities management: A literature review and future research directions - ScienceDirect[J]. Journal of Building Engineering, 24(C): 100755-100755.

Caption of figures

The caption of all figures is still far too short. The captions must clearly describe the figures.

Description and explanation of figures in text

All figures must be desribed and explained in the text. Often, just the short caption is repeated in the text.

Open questions

The term "Trait Based Object" must be explained. And it must be explained why the use of TBO is important to the research.

Lines 217-218: "The lightweight of geometric information ..." 
It is unclear what the autors mean ... maybe "Reducing geometric information".

Figure 7:  This is one example of missing explanation of the figures.
The only description of this figure is "The parsing process is shown in the figure 7." But the logic of the figure is not clear ... hence, just a repetition of the (too short) caption. It needs to be better explained.

Qualtiy of English Language and Style of Document

There are several mistakes which should not happen in a 2nd review round.

Grammar errors: In several cases, the wrong tense is used in a sentence. The authors use the past tense where the present tense is appropriate.

Several times wrong words are used (e.g., examine, disposal).
Maybe the authors mean "to train" instead of "to examine".
Maybe the authors mean "rescue" or "escape" instead of "disposal".

Some words begin with a lower case letter, although a capital letter is required. And sometimes a capital letter is used although a lower case letter is the right choice.

Several times words or terms occur twice (in a row).

Words are still misspelled (e.g., algoxrithm).

Too often, spaces are either in a place where they do not belong (before a punctuation mark) or they are missing (after a punctuation mark or between a word and a number).

Chapter 2, in lines 189 to 200 the line space is reduced. 

Chapter 3, starting line 435 the font size is reduced.

Author Response

We thank the Editor-in-Chief for allowing us to revise this manuscript. We also greatly appreciate the thorough and thoughtful evaluation of our paper by the anonymous reviewers.  Their insightful comments helped us significantly improve the quality of this paper.

Following the editor and the reviewers’ advice, we have undertaken a minor revision of this manuscript. In the files of revision with changes marked, the yellow highlights are changes for responses to the editor and reviewer comments.
